# Cluster Attention for Graph Machine Learning

## Abstract

Message passing neural networks have recently become the most popular approach to graph machine learning tasks, however, their receptive field is limited by the number of message-passing layers. To increase the receptive field, using graph transformers with global attention has been proposed, however, global attention does not take into account the graph topology and thus lacks graph-structure-based inductive biases which are typically very important for graph machine learning tasks. In this work, we propose an alternative approach: cluster attention (CLATT). We divide graph nodes into clusters with off-the-shelf graph community detection algorithms and let each node attend to all other nodes in each cluster. CLATT provides large receptive fields while still having strong graph-structure-based inductive biases. We show that augmenting message-passing neural networks or graph transformers with CLATT significantly improves their performance on a wide range of graph datasets including datasets from the recently introduced GraphLand benchmark representing real-world applications of graph machine learning.

## 1 Introduction

Graphs are a natural way to represent data from different domains such as social networks (both real-life and virtual), computer networks, transportation networks, co-purchasing networks, molecules, connectomes, neural network architectures, various physical and biological systems, or even interconnected abstract concepts. Thus, machine learning on graph-structured data, henceforth referred to as Graph Machine Learning (GML), has attracted a lot of attention recently. In particular, Graph Neural Networks (GNNs) — a class of models that first appeared a long time ago (Sperduti & Starita, 1997; Gori et al., 2005; Scarselli et al., 2008) but has been refined more recently (Duvenaud et al., 2015; Kipf & Welling, 2017; Hamilton et al., 2017) — have become the most popular models for most GML tasks in the past decade. Most modern GNNs can be unified by the Message Passing Neural Networks (MPNNs) framework (Gilmer et al., 2017) in which each node in the graph in each neural network layer sends messages to its neighbors and aggregates incoming messages to form its new representation based on them. The graph-structure based relational inductive biases of MPNNs which send messages along graph edges turned out to be very effective for modeling many real-world networks (Battaglia et al., 2018). However, in each MPNN layer, the model only exchanges information between neighboring nodes. Thus, the model's receptive field, i.e., the maximum shortest path distance information can travel between nodes in the graph, is upper bounded by the number of message-passing layers. This has been argued to be a limitation of MPNNs as it prevents taking into account long-range interaction between nodes that might happen in some networks. Thus, graph transformers with their all-to-all attention have been proposed as an alternative neural model architecture for GML (Kreuzer et al., 2021; Ying et al., 2021; Rampášek et al., 2022), inspired by the success of the transformer architecture in the field of natural language processing (Vaswani et al., 2017). However, all-to-all attention does not have any information about the graph structure which is arguably very important for solving GML tasks and thus has to be integrated into the model in some other ways (typically via positional encodings or MPNN-transformer hybridization).

It has been observed that many real-world networks exhibit community structure — presence of well-defined clusters of densely connected nodes with sparser connections between these clusters (Girvan & Newman, 2002). The problem of finding such clusters — graph clustering — has been extensively studied in network science and machine learning communities, and many methods for

it have been proposed over the years (Fortunato, 2010). Such algorithms can be used to divide graph nodes into clusters where nodes within the same cluster are strongly related, where the exact definition of this relation and thus the meaning of the obtained clusters can differ between different clustering algorithms.

In this work, we argue that various partitions of a graph into node clusters provide very useful information about the graph structure and thus it can be beneficial to integrate these partitions into GML models. Specifically, we propose *cluster attention (CLATT)* — a graph-based attention mechanism in which nodes can attend to other nodes in the same cluster and thus exchange information with them. We view cluster attention as a middle ground between MPNNs in which only neighboring nodes can interact with each other and graph transformers in which all nodes can interact with each other. Cluster attention allows capturing longer-ranged dependencies than MPNNs while still providing strong graph-structure-based inductive biases which graph transformers lack.

We argue that the graph-structure-based inductive biases of cluster attention are complementary to those of MPNNs and thus propose augmenting MPNNs with cluster attention. Similarly, graph transformers can be augmented with CLATT to provide these global attention models with stronger graph-structure-based inductive biases. In experiments on a diverse set of 11 real-world graph datasets we demonstrate that cluster attention can significantly improve the performance of both MPNNs and graph transformers.

The rest of this paper is structured as follows. Section 2 provides the necessary background on GML with neural networks and on graph node clustering. In Section 3, we formally define cluster attention and discuss its implementation details. In Section 4, we discuss our approach to selecting graph node clustering algorithms to use with cluster attention. Section 5 presents our experimental results. In Section 6, we discuss the limitations of cluster attention. Section 7 provides concluding remarks.

## 2 BACKGROUND AND RELATED WORK

### 2.1 GRAPH NEURAL NETWORKS

In the recent years, Graph Neural Networks (GNNs), in particular Message Passing Neural Networks (MPNNs) (Gilmer et al., 2017), have become the most dominant approach to graph machine learning tasks. In each message-passing layer of MPNNs, each node aggregates representations (messages) from its neighbors in the graph an updates its own representation based on them. Many GNN architectures falling under the MPNNs framework have been proposed (Kipf & Welling, 2017; Hamilton et al., 2017; Veličković et al., 2018; Xu et al., 2019), mostly differing in their aggregation function. The receptive field of MPNNs, i.e., the maximum distance in the graph at which nodes can interact with each other, is equal to the number of message-passing layers. This is often viewed as a limitation, since long-range dependencies can exist in some graph machine learning problems. While there have been some works attempting to increase the receptive field of MPNNs (Abu-El-Haija et al., 2019; Finder et al., 2025), a different and more radical approach has become more popular: replacing MPNNs with models with all-to-all Transformer-style attention (Vaswani et al., 2017) that has global receptive field. Such models became known as Graph Transformers (GTs) (Kreuzer et al., 2021; Ying et al., 2021; Rampášek et al., 2022). However, global attention has no information about graph topology and thus this approach foregoes graph-structure-based inductive biases of MPNNs and essentially handles the data as a set rather than a graph. To inject graph information into GTs, various positional encoding (PEs) are typically used that are added to node features or attention weights. Such PEs are often based on Laplacian eigenvectors (Dwivedi et al., 2020), graph distances (Ying et al., 2021), or random walks (Dwivedi et al., 2022). However, despite the success of the Transformer architecture in other fields such as natural language processing, computer vision, and audio processing, and despite considerable research efforts to adapt Transformers to graph machine learning, recent benchmarking works show that GTs tend to not provide advantages over MPNNs (Tönshoff et al., 2023; Luo et al., 2024; 2025).

It is also important to note that there are actually two types of models commonly referred to as Graph Transformers. Besides Graph Transformers with all-to-all attention discussed above, another type consists of models that adopt the general Transformer architecture to GNNs but limit attention to each node's neighborhood, i.e., each node can only attend to its neighbors (Shi et al., 2021; Dwivedi

& Bresson, 2021). Such models fit into the MPNN framework and have been shown to be a very strong MPNN variant (Platonov et al., 2023b; Bazhenov et al., 2025). To distinguish between these two model classes, we will henceforth refer to models with all-to-all attention as Global Graph Transformers (GGTs) and to models with neighborhood attention as Local Graph Transformers (LGTs).

## 2.2 GRAPH CLUSTERING

Graph node clustering (henceforth graph clustering), also referred to as community detection, graph partitioning, and modular structure inference, is an important and long-studied problem in network science and graph machine learning with many applications in various fields. Many widely different approaches to graph clustering have been proposed, see Fortunato (2010); Fortunato & Hric (2016); Schaeffer (2007) for detailed overviews. Here we briefly review some of the most well-known approaches. Many methods are based on intuitive heuristics such as iterative propagation of influence between nodes (Raghavan et al., 2007) or divisive clustering by removing "bridge-like" edges based on betweenness centrality (Girvan & Newman, 2002) or approximate minimum cuts (and variants such as ratio cuts and normalized cuts) (Fiedler, 1973; Pothen et al., 1990; Shi & Malik, 2000; Meila & Shi, 2000; Andersen et al., 2007; Riolo & Newman, 2014). An approach that gained a lot of popularity is maximizing a certain measure (quality function) that shows how well-connected are the nodes within a single cluster compared to nodes from different clusters. The most well-known such measure is modularity (Newman & Girvan, 2004; Newman, 2006b), and many algorithms to maximize it have been proposed (Clauset et al., 2004; Duch & Arenas, 2005; Reichardt & Bornholdt, 2006; Guimera & Nunes Amaral, 2005; Newman, 2006a), the most popular one being the greedy Louvain algorithm (Blondel et al., 2008). However, modularity maximization for graph clustering has some limitations (Fortunato & Hric, 2016), including a problem known as the resolution limit (Fortunato & Barthelemy, 2007). An alternative to modularity measure that does not suffer from the resolution limit is the Constant Potts Model (CPM) (Traag et al., 2011), which can also be minimized with the Louvain algorithm. An improvement to the Louvain algorithm that can be used to greedily maximize either modularity or CPM is the Leiden algorithm (Traag et al., 2019). Another popular approach to graph clustering is based on statistical inference: the graph is assumed to be generated from a certain random graph model family parametrized by community assignments of nodes, and the assignment with maximum likelihood is searched for. Methods following this approach differ in what model family they assume and how they maximize likelihood (Hastings, 2006; Newman & Leicht, 2007; Zanghi et al., 2008; Hofman & Wiggins, 2008; Copic et al., 2009; Karrer & Newman, 2011; Peixoto, 2014a;b; Zhang et al., 2015; Prokhorenkova & Tikhonov, 2019; Zhang & Peixoto, 2020). The approaches discussed above consider unattributed graphs, i.e., graphs without node features. More recently, there appeared methods for clustering attributed graphs with GNNs considering both graph structure and node features (Tsitsulin et al., 2023).

## 3 CLUSTER ATTENTION

### 3.1 METHOD

The standard MPNN mechanism of sending messages along graph edges provides the model with strong graph-structure-based inductive biases, but prevents nodes further away in the graph than the number of message-passing layers from exchanging information. We would like to design a mechanism that allows longer-ranged interactions while still being rooted in the graph structure and thus providing useful inductive biases to the model (in contrast to the all-to-all attention of GGTs). We propose using graph clustering to achieve this. Specifically, we consider graph clustering as a preprocessing step and then allow nodes within the same cluster to interact with each other through attention. Essentially, we use all-to-all attention on the level of individual clusters rather than on the full-graph level. Such an attention mechanism allows capturing longer-ranged dependencies than the classic message-passing mechanism while still limiting which nodes can interact with each other based on the graph structure. We call this mechanism *Cluster Attention (CLATT)*. We propose using CLATT together with classic message passing since they provide different kinds of graph-structure-based inductive biases that can be beneficial to each other. This is reminiscent of how GGTs are often combined with classic message passing (Rampášek et al., 2022). Note that stacking multiple layers with message passing and CLATT allows even nodes from different clusters to interact with

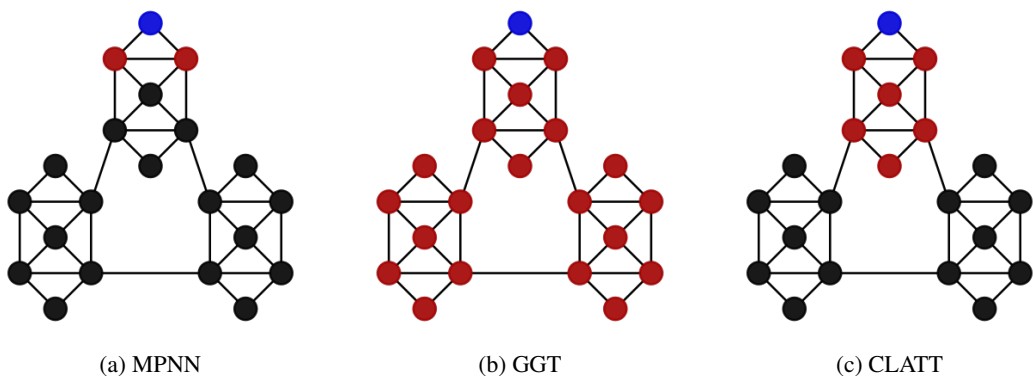

(a) MPNN  (b) GGT  (c) CLATT

Figure 1: Receptive fields of a single layer of different GML models. The blue node can interact with the red nodes. In an MPNN, the receptive field is based on the graph structure, but only neighboring nodes can interact with each other. GGT allows all nodes to interact with each other, but completely disregards the graph structure. CLATT allows nodes within the same cluster to interact, which provides longer-ranged interactions than in an MPNN, but still relies on the graph structure to disallow most possible pairwise interactions.

each other thus allowing the model to capture even longer-ranged dependencies. In Figure 1, we provide a simple example of how the receptive fields of a single layer of an MPNN, a GGT, and CLATT differ.

Note that modern efficient graph clustering algorithms such as the Leiden algorithm (Traag et al., 2019) run orders of magnitude faster than it takes to train a GNN, thus the graph clustering preprocessing step takes negligible time. However, a question remains: how to choose a graph clustering algorithm for CLATT from dozens different approaches that have been proposed in the literature. We will discuss clustering algorithm selection in detail in Section 4, but for now let us note that it is not necessary to limit CLATT to *a single* graph clustering algorithm. We can select *multiple* graph clusterings obtained with different algorithms, apply CLATT to each one, and then concatenate the resulting embeddings for each node. Different clustering algorithms rely on different assumptions and have different inductive biases, thus producing significantly different clustering that capture different types of information about the graph structure, and using multiple such clusterings lets a model access all of these different information.

Now let us describe CLATT more formally. Let $G = (V, E)$ be a graph with the nodeset $V$ and the edgeset $E$. Let $\mathcal{C}$ be an ordered set of clusterings that contains one or more clusterings that we want to use with CLATT. For each clustering $C \in \mathcal{C}$ and each node $i \in V$ we will denote by $C(i)$ the cluster (the set of nodes) to which the node $i$ belongs in the clustering $C$. Let $\mathbf{x}_i \in \mathbb{R}^d$ be the representation of node $i \in V$ before the CLATT operation, i.e., the input to CLATT, and let $\mathbf{y}_i \in \mathbb{R}^{d|\mathcal{C}|}$ be the representation of node $i \in V$ after the CLATT operation, i.e., the output of CLATT. The complete CLATT output representation $\mathbf{y}_i \in \mathbb{R}^{d|\mathcal{C}|}$ for node $i$ is the concatenation of CLATT output representations for each clustering $C \in \mathcal{C}$:

$$\mathbf{y}_i = \text{concatenate}\left(\mathbf{y}_i^C \text{ for } C \text{ in } \mathcal{C}\right) \tag{1}$$

The CLATT output representation $\mathbf{y}_i^C$ for a single clustering $C \in \mathcal{C}$ for node $i$ is computed via attention between nodes belonging to the cluster $C(i)$ in the following way (for ease of notation we assume a single attention head, but multihead attention can be used instead):

$$\mathbf{y}_i^C = \sum_{j \in C(i)} p_{ij}^C \mathbf{v}_j^C, \tag{2}$$

where $p_{ij}^C \in [0, 1]$ is the weight of attention from node $i$ to node $j$ obtained with softmax as

$$p_{ij}^C = \frac{\exp\left(\alpha_{ij}^C\right)}{\sum_{j \in C(i)} \exp\left(\alpha_{ij}^C\right)}, \tag{3}$$

$\alpha_{ij}^C$ is the pre-softmax logit of attention from node $i$ to node $j$ obtained as a scaled dot product between the corresponding query and key vectors:

$$\alpha_{ij}^C = \frac{\langle \mathbf{q}_i^C, \mathbf{k}_j^C \rangle}{\sqrt{d}}, \tag{4}$$

and $\mathbf{q}_i^C, \mathbf{k}_i^C, \mathbf{v}_i^C \in \mathbb{R}^d$ are the query, key, and value vectors respectively for node $i \in V$ obtained as learnable linear transformations of the node's input representation:

$$\mathbf{q}_i^C = \mathbf{W}_q^C \mathbf{x}_i + \mathbf{b}_q^C, \qquad \mathbf{k}_i^C = \mathbf{W}_k^C \mathbf{x}_i + \mathbf{b}_k^C, \qquad \mathbf{v}_i^C = \mathbf{W}_v^C \mathbf{x}_i + \mathbf{b}_v^C, \tag{5}$$

where $\mathbf{W}_q^C, \mathbf{W}_k^C, \mathbf{W}_v^C \in \mathbb{R}^{d \times d}$ and $\mathbf{b}_q^C, \mathbf{b}_k^C, \mathbf{b}_v^C$ are learnable parameters. The complete CLATT output representation $\mathbf{y}_i \in \mathbb{R}^{d|\mathcal{C}|}$ for node $i$ is then concatenated with the message-passing output for node $i$ (which is typically of dimension $d$) and passed through a learnable linear layer that reduces its dimension back to $d$.

### 3.2 IMPLEMENTATION DETAILS

CLATT can be naively implemented by adding edges connecting all pairs of nodes within the same cluster to the graph. However, this will likely create a very dense graph for which graph attention operation meant for sparser graphs will be inefficient. Instead, we transform the tensor of node representation of shape `num_nodes × hidden_dim` to a padded tensor of shape `num_clusters × max_cluster_size × hidden_dim` and apply standard dense attention along the `max_cluster_size` dimension with a mask that blocks attention to padding. Modern efficient attention implementations[1] combined with JIT-compilation[2] make this operation very efficient. Further, some modern frameworks provide so-called "ragged / jagged" tensors[34] which can be used to stack sequences of different length into a single tensor-like structure, which in our case can allow avoiding padding and make CLATT implementation even more efficient.

## 4 CLUSTERING ALGORITHMS SELECTION

Considering the wide and diverse range of graph clustering algorithms available (see Section 2.2), it is important to choose appropriate clustering algorithms to use with CLATT. Different clustering algorithms rely on different principles and make different assumptions about the data, and thus can produce significantly different results. Since real-world graphs come from many different applications and have very different structures, the best clustering algorithm for use with CLATT can also be different for different datasets and tasks. First, we note that we do not need to limit CLATT to using *a single* clustering: we can select *multiple* clustering algorithms, obtain a node clustering from each one of them, apply CLATT to each of these clusterings, and combine the results (in our work, we simply concatenate the embeddings produced by applying CLATT to different clusterings). However, we still need to choose a relatively small set of clustering algorithms to use from the wide range of available ones. In this selection, we use the following two criteria: first, we only consider efficient algorithms that can be run relatively fast on large-scale real-world graphs, making graph clustering a cheap preprocessing step compared to GNN training time; second, we aim to select a set of clustering algorithms that produce very different results compared to each other,

---

[1]https://docs.pytorch.org/docs/stable/generated/torch.nn.functional.scaled_dot_product_attention.html
[2]https://docs.pytorch.org/tutorials/intermediate/torch_compile_tutorial.html
[3]https://docs.pytorch.org/docs/stable/nested.html
[4]https://docs.pytorch.org/tutorials/unstable/nestedtensor.html

thus providing CLATT with very different information about the graph structure when used together. Based on these criteria, we select 4 clustering algorithms. In practice, different clustering algorithms can have different usefulness for different datasets and tasks, and there is no need to always use all 4 of them. Thus, we treat the choice of clustering algorithm as a hyperparameter. First, we train a model 4 times with each clustering individually, then we take those of the considered clusterings that improved results on the validation set and use them together for the main training run. To speed up experiments, we only run this procedure with one of the considered models (specifically, LGT) and use the same selection of clusterings for all other models (we assume that the selection of clusterings will transfer well between different models, but it is possible that even better results can be achieved if clustering selection is run for each model individually).

Now, let us describe the 4 clustering algorithms that we use.

- First, we consider the widely used Leiden algorithm (Traag et al., 2019) — an improvement of the popular Louvain algorithm (Blondel et al., 2008). Leiden algorithm greedily optimizes a measure of how well-connected are nodes within a single cluster compared to nodes from different clusters. As such a measure, we use the Constant Potts Model (CPM) (Traag et al., 2011), which does not suffer from the resolution limit problem (Fortunato & Barthelemy, 2007), in contrast to the more widely use modularity measure (Newman & Girvan, 2004; Newman, 2006b). We use the official algorithm implementation.[5]

- Next, we consider a statistical clustering algorithm. Specifically, we use an algorithm that fits a Bayesian planted partition model by Zhang & Peixoto (2020). We use the official algorithm implementation from the `graph-tool` library (Peixoto, 2017).

- Note that the two algorithms discussed above assume *assortative (homophilous)* community structure, i.e., nodes within the same cluster are more likely to be connected than nodes from different clusters. While such community structure is natural, sometimes it can also be useful to find *disassortative (heterophilous)* clusters, i.e., clusters of nodes that are not necessarily connected with each other but share the same structural role in the graph. For example, when a lot of leaf nodes are connected to one node in the graph, they can be expected to share some properties, and it can be useful to put them in a cluster of their own even though they are not connected with each other. Thus, we also consider a statistical clustering algorithm that does not make the assumption of cluster assortativity and can detect disassortative clusters. Specifically, we use an algorithm by Peixoto (2014b). This algorithm produces a hierarchical clustering, however, we only consider the level of the hierarchy with the smallest nontrivial number of clusters as we find that other levels typically find clusters of very small size. We use the official algorithm implementation from the `graph-tool` library.

- Above, we have described 3 very different graph clustering algorithms. However, we find that it can sometimes be useful to also consider the *feature similarity* of nodes, i.e., ignore the graph structure and apply a classic clustering algorithm to node features. This approach will allow the model to exchange information between nodes that have similar features no matter how far they are in the graph, which can be useful for some applications. For this purpose, we use the classic k-means algorithm (Forgy, 1965; Lloyd, 1982). We use the implementation from the `scikit-learn` library (Pedregosa et al., 2011). However, since the k-means algorithm relies on euclidean distances it will not produce good clusterings for datasets with features of very different scales, which are common in practical applications. Thus, instead of directly clustering node features, we train an auxiliary ResMLP model (a simple graph-agnostic model that is very fast to train) and cluster its hidden representations of graph nodes instead.

The 4 considered algorithms are very efficient and also typically produce significantly different clusterings, thus satisfying our desiderata. To verify that these algorithm produce significantly different results, we can compare the clusterings obtained from these algorithms using some *clustering similarity measure*. While many such measures have been proposed in the literature (Gösgens et al., 2021), we use the measure known as *Correlation Coefficient (CC)* since it has been shown to satisfy many desirable properties (Gösgens et al., 2021). We show the similarity matrices of the clusterings obtained by the 4 considered algorithms on 4 datasets used in our experiments (see Appendix A for dataset descriptions) in Figure 2. Indeed, the similarity values of clusterings are typically relatively low, with the graph-agnostic k-means clusterings being particularly dissimilar to the graph-based clusterings, as expected.

---

[5]`leidenalg.readthedocs.io`

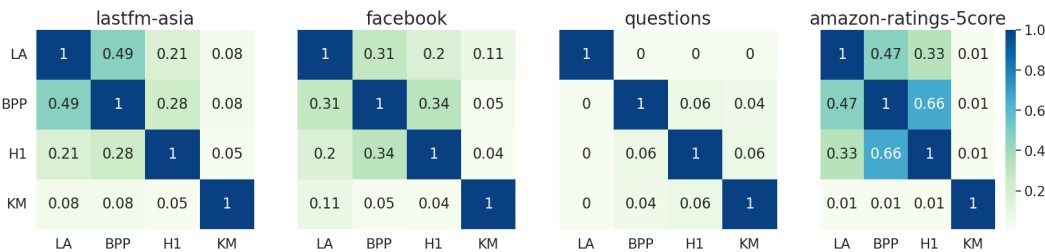

Figure 2: Similarities of clusterings produced by different node clustering algorithms on 4 different datasets. Correlation Coefficient similarity measure is reported. LA is Leiden algorithm, BPP is Bayesian planted partition model, H1 is the highest nontrivial level of hierarchical statistical clustering, KM is k-means clustering of node representations.

Table 1: Characteristics of the considered graph datasets.

| | node regression | | | node classification | | | | | | | |
|---|---|---|---|---|---|---|---|---|---|---|---|
| | city-roads-M | city-roads-L | avazu-ctr | tolokers-2 | hm-categories | pokec-regions | questions | lastfm-asia | facebook | amazon-r-5core | amazon-r-full |
| # nodes | 57.1K | 142.3K | 76.3K | 11.8K | 46.5K | 1.6M | 48.9K | 7.6K | 22.5K | 24.5K | 334.9K |
| # edges | 107.1K | 231.6K | 11.0M | 519.0K | 10.7M | 22.3M | 153.5K | 27.8K | 170.8K | 93.1K | 925.8K |
| avg degree | 3.75 | 3.26 | 288.04 | 88.28 | 460.92 | 27.32 | 6.28 | 7.29 | 15.20 | 7.60 | 5.53 |
| median degree | 4 | 3 | 71 | 30 | 45 | 13 | 1 | 4 | 7 | 5 | 4 |
| avg distance | 126.75 | 194.05 | 3.55 | 2.79 | 2.45 | 4.68 | 4.29 | 5.23 | 4.97 | 16.24 | 11.95 |
| diameter | 383 | 553 | 14 | 11 | 13 | 14 | 16 | 15 | 15 | 46 | 47 |
| global clustering | 0.00 | 0.00 | 0.24 | 0.23 | 0.27 | 0.05 | 0.02 | 0.18 | 0.23 | 0.32 | 0.21 |
| avg local clustering | 0.00 | 0.00 | 0.85 | 0.53 | 0.70 | 0.11 | 0.03 | 0.22 | 0.36 | 0.58 | 0.40 |
| degree assortativity | 0.70 | 0.74 | −0.30 | −0.08 | −0.35 | 0.00 | −0.15 | 0.02 | 0.08 | −0.09 | −0.06 |
| # classes | N/A | N/A | N/A | 2 | 21 | 183 | 2 | 18 | 4 | 5 | 5 |
| unbiased homophily | N/A | N/A | N/A | 0.10 | 0.38 | 0.98 | 0.06 | 0.97 | 0.90 | 0.28 | 0.22 |
| target assortativity | 0.74 | 0.72 | 0.18 | N/A | N/A | N/A | N/A | N/A | N/A | N/A | N/A |

## 5 EXPERIMENTS

### 5.1 DATASETS

In our experiments, we aim to show that CLATT can improve results on diverse datasets from different domains. In particular, we aim to go beyond the classic citation network datasets, extensive reliance on which has been criticized recently (Bechler-Speicher et al., 2025; Bazhenov et al., 2025), and use graph datasets from more realistic applications. Thus, we select 11 datasets from different fields and with different tasks, showcasing a wide range of graph sizes and structural characteristics. First, we use 6 datasets from the recently introduced GraphLand benchmark (Bazhenov et al., 2025) that focuses on graph datasets from realistic industrial applications and with rich node features. Specifically, we use road networks city-roads-M and city-roads-L, a networks of internet-connected devices avazu-ctr, a network of crowdsourcing platform workers tolokers-2, a co-purchasing network hm-categories, and a social network pokec-regions. For datasets from GraphLand, we use the official RL (random low) 10%/10%/80% train/val/test data splits. Further, we use two classic social network datasets lastfm-asia (Rozemberczki & Sarkar, 2020) and facebook (Rozemberczki et al., 2019). For these two datasets, we use random 10%/10%/80% train/val/test data splits. Then, we use two datasets from Platonov et al. (2023b): co-purchasing network amazon-ratings and question-answering network questions. For these two datasets, we use random 50%/25%/25% train/val/test data splits. Finally, we notice that the amazon-ratings dataset has a peculiar graph structure due to being downsampled by taking the 5-core (Malliaros et al., 2020) of the full graph. For this reason, we further refer to this dataset as amazon-ratings-5core, and we also prepare a new dataset which is the full version of the same co-purchasing network, which we call amazon-ratings-full. Besides being more than an order of magnitude larger, this dataset exhibits very different struc-

Table 2: Experimental results. $R^2$ is reported for regression datasets, Average Precision is reported for binary classification datasets, Accuracy is reported for multiclass classification datasets. TL means time limit exceeded (24 hours for a single model run). For each pair consisting of a base model and its version augmented with CLATT, we highlight the best of the two results with bold and additionally underline it if the difference between the two results is statistically significant.

(a) Experimental results for GraphLand datasets under the `RL` data split.

|  | regression | | | bin. class. | mult. class. | |
| --- | --- | --- | --- | --- | --- | --- |
|  | city-roads-M | city-roads-L | avazu-ctr | tolokers-2 | hm-categories | pokec-regions |
| GCN | $59.05 \pm 0.16$ | $53.26 \pm 0.14$ | $30.47 \pm 0.27$ | $51.48 \pm 0.81$ | $61.88 \pm 0.23$ | $34.99 \pm 0.17$ |
| GCN-CLATT | $\underline{\mathbf{60.18 \pm 0.30}}$ | $\underline{\mathbf{55.40 \pm 0.14}}$ | $\underline{\mathbf{31.06 \pm 0.37}}$ | $\underline{\mathbf{54.09 \pm 0.58}}$ | $\underline{\mathbf{65.69 \pm 0.53}}$ | $\underline{\mathbf{47.45 \pm 0.34}}$ |
| GraphSAGE | $57.51 \pm 0.53$ | $52.43 \pm 0.25$ | $31.84 \pm 0.24$ | $53.87 \pm 0.78$ | $56.72 \pm 0.31$ | $37.80 \pm 0.35$ |
| GraphSAGE-CLATT | $\underline{\mathbf{58.69 \pm 0.61}}$ | $\underline{\mathbf{55.43 \pm 0.23}}$ | $\underline{\mathbf{32.55 \pm 0.22}}$ | $\underline{\mathbf{54.76 \pm 0.63}}$ | $\underline{\mathbf{62.12 \pm 0.28}}$ | $\underline{\mathbf{48.60 \pm 1.03}}$ |
| LGT | $58.05 \pm 0.58$ | $53.38 \pm 0.12$ | $30.87 \pm 0.47$ | $55.70 \pm 0.28$ | $69.25 \pm 0.25$ | $46.41 \pm 0.20$ |
| LGT-CLATT | $\underline{\mathbf{60.05 \pm 0.31}}$ | $\underline{\mathbf{55.58 \pm 0.29}}$ | $\underline{\mathbf{31.87 \pm 0.26}}$ | $\underline{\mathbf{56.75 \pm 0.34}}$ | $\underline{\mathbf{70.25 \pm 0.35}}$ | $\underline{\mathbf{49.18 \pm 0.89}}$ |
| GGT-DW | $53.14 \pm 0.50$ | $48.24 \pm 0.82$ | $27.78 \pm 0.62$ | $56.27 \pm 0.34$ | $41.72 \pm 0.41$ | TL |
| GGT-DW-CLATT | $\underline{\mathbf{53.99 \pm 0.68}}$ | $\underline{\mathbf{49.65 \pm 0.38}}$ | $\underline{\mathbf{28.74 \pm 0.48}}$ | $\underline{\mathbf{57.93 \pm 0.45}}$ | $\underline{\mathbf{55.31 \pm 0.24}}$ | TL |
| GGT-Lap | $54.32 \pm 0.64$ | $46.48 \pm 0.99$ | $24.27 \pm 1.16$ | $49.46 \pm 1.05$ | $37.79 \pm 0.34$ | TL |
| GGT-Lap-CLATT | $\underline{\mathbf{56.80 \pm 0.20}}$ | $\underline{\mathbf{51.75 \pm 0.14}}$ | $\underline{\mathbf{26.52 \pm 0.34}}$ | $\underline{\mathbf{53.78 \pm 0.83}}$ | $\underline{\mathbf{53.45 \pm 0.15}}$ | TL |

(b) Experimental results for other datasets.

|  | bin. class. | mult. class. | | | |
| --- | --- | --- | --- | --- | --- |
|  | questions | lastfm-asia | facebook | amazon-ratings-5core | amazon-ratings-full |
| GCN | $18.60 \pm 0.59$ | $80.62 \pm 0.36$ | $91.00 \pm 0.14$ | $49.65 \pm 0.50$ | $39.28 \pm 0.13$ |
| GCN-CLATT | $\underline{\mathbf{21.13 \pm 0.42}}$ | $\underline{\mathbf{84.46 \pm 0.27}}$ | $\underline{\mathbf{91.43 \pm 0.27}}$ | $\underline{\mathbf{52.89 \pm 0.30}}$ | $\underline{\mathbf{39.60 \pm 0.12}}$ |
| GraphSAGE | $20.41 \pm 0.47$ | $80.90 \pm 0.46$ | $91.22 \pm 0.21$ | $\mathbf{54.94 \pm 0.38}$ | $38.73 \pm 0.12$ |
| GraphSAGE-CLATT | $20.91 \pm 0.29$ | $\underline{\mathbf{84.87 \pm 0.21}}$ | $\underline{\mathbf{91.70 \pm 0.13}}$ | $54.62 \pm 0.26$ | $\underline{\mathbf{39.22 \pm 0.06}}$ |
| LGT | $17.86 \pm 0.52$ | $83.97 \pm 0.27$ | $92.17 \pm 0.20$ | $51.26 \pm 0.38$ | $39.03 \pm 0.08$ |
| LGT-CLATT | $\underline{\mathbf{22.04 \pm 0.43}}$ | $\underline{\mathbf{84.68 \pm 0.29}}$ | $\underline{\mathbf{92.98 \pm 0.10}}$ | $\underline{\mathbf{53.75 \pm 0.36}}$ | $\underline{\mathbf{39.40 \pm 0.10}}$ |
| GGT-DW | $16.17 \pm 0.87$ | $78.69 \pm 0.50$ | $86.84 \pm 0.27$ | $46.22 \pm 0.79$ | TL |
| GGT-DW-CLATT | $\underline{\mathbf{19.90 \pm 1.25}}$ | $\underline{\mathbf{82.61 \pm 0.53}}$ | $\underline{\mathbf{90.59 \pm 0.25}}$ | $\underline{\mathbf{48.34 \pm 0.57}}$ | TL |
| GGT-Lap | $18.15 \pm 0.86$ | $63.50 \pm 1.16$ | $73.73 \pm 0.73$ | $45.97 \pm 0.71$ | TL |
| GGT-Lap-CLATT | $\underline{\mathbf{21.32 \pm 0.37}}$ | $\underline{\mathbf{82.51 \pm 0.31}}$ | $\underline{\mathbf{89.21 \pm 0.34}}$ | $\underline{\mathbf{52.66 \pm 0.30}}$ | TL |

tural characteristics compared to `amazon-ratings-5core`. For this dataset, we use a random 10%/10%/80% train/val/test data split. A detailed description of this new dataset and how it differs from `amazon-ratings-5core` is provided in Appendix C.

To showcase the diversity of graph structures in the considered datasets, we provide some of their structural characteristics in Table 1. More details on these datasets and their characteristics are provided in Appendix A. For example, note that we use a range of both homophilous datasets (`city-roads-M`, `city-roads-L`, `pokec-regions`, `lastfm-asia`, `facebook`) and non-homophilous datasets (`avazu-ctr`, `tolokers-2`, `hm-categories`, `questions`, `amazon-ratings-5core`, `amazon-ratings-full`) to demonstrate that CLATT works well for both of these types of datasets.

## 5.2 MODELS

As discussed above, we propose integrating CLATT into MPNNs to combine the benefits of local message passing and more long-ranged cluster attention. We combine CLATT with three different MPNN models and show that CLATT improvers results for all of them. Specifically, we use two classic GNNs — GCN (Kipf & Welling, 2017) and GraphSAGE (Hamilton et al., 2017) — and also a Local Graph Transformer (LGT), which has been shown to often provide stronger results in recent benchmarking works (Platonov et al., 2023b; Bazhenov et al., 2025). For all models, we use the modifications from Platonov et al. (2023b) which augment models with skip-connections (He et al., 2016), layer normalization (Ba et al., 2016), and MLP blocks between message-passing

layers. These modifications have been shown to significantly improve the results of MPNNs (Luo et al., 2024; 2025).

Further, we also combine CLATT with Global Graph Transformer (GGT) and show that it improves performance in this case as well. We experiment with two versions of GGT that differ by the positional encodings used: we use either classic DeepWalk node embeddings (Perozzi et al., 2014) or Laplacian eigenvectors (Dwivedi et al., 2020; Belkin & Niyogi, 2001). We denote these models as GGT-DW and GGT-Lap, respectively. Note that while many alternative node embedding methods have been proposed since DeepWalk, it has been shown that in practice DeepWalk still provides some of the best results (Gurukar et al., 2022).

For all models, we perform extensive hyperparamter search. The details of it, as well as other information about our experimental setting, are provided in Appendix B.

### 5.3 EXPERIMENTAL RESULTS

We report the results of our experiments in Table 2. For each of the considered models, we compare the original version with its version augmented with CLATT (which is denoted with -CLATT suffix). We can see that CLATT improves the performance of the base models in all cases with the only exception of the GraphSAGE model on the `amazon-ratings-5core` dataset, and these improvements are often quite substantial. For example, on the `pokec-region` dataset, CLATT improves the performance of GCN and GraphSAGE by 12 and 10 percentage points, respectively. The improvements from CLATT are not limited to MPNNs: for GGTs, CLATT always improves performance by providing these models with graph-structure-based inductive bias. For example, on the `hm-categories` dataset, CLATT improves the performance of GGT-DW and GGT-Lap by more than 13 and 15 percentage points, respectively.

We note that CLATT brings improvements on datasets with very different graph structural characteristics. As discussed above, the considered datasets cover both homophilous and non-homophilous graphs. Further, CLATT improves model performance on both very dense graph (`avazu-ctr`, `hm-categories`) and very sparse graphs (`city-roads-M`, `city-roads-L`, `questions`, `amazon-ratings-full`). Note that `city-roads-M` and `city-roads-L` datasets, representing transportation networks, do not exhibit well-defined clusters (as can be expected given their zero clustering coefficients and very large average distances), in contrast to, for example, social networks, yet CLATT brings strong benefits on them as well. More details on the diversity of graph structural characteristics covered by the considered datasets are provided in Appendix A.

## 6 LIMITATIONS

Graphs can be used to represent data from very different domains, and thus can exhibit a wide range of structural characteristics and relationships between graph structure, node features, and node labels. Thus, we do not expect that a single method will perform well on all realistic graphs. Yet, we show that CLATT can improve the performance of both MPNNs and GGTs (two currently most popular families of neural architectures for graph machine learning) on a wide range of graph datasets from real-world applications of graph machine learning. However, of course counterexamples of graphs on which CLATT does not improve performance are possible. We expect that the effectiveness of CLATT is correlated with the degree of the presence of meaningful cluster structure in the graph. While many graphs exhibit easily detectable cluster structure, some graphs, for example, highly regular grid-like graphs, are unlikely to contain meaningful clusters and thus probably will not benefit from CLATT. However, many real-world networks do contain useful cluster structure that CLATT can leverage to improve model performance, which we demonstrate in our experiments.

## 7 CONCLUSION

In this work, we introduce a new technique for graph machine learning — cluster attention (CLATT), which can be viewed as a middle ground between local message-passing and global attention that incorporates strong graph-structure-based inductive biases while allowing for long-range interactions. We show that CLATT significantly improves results of both MPNNs and GGTs on a wide range of diverse real-world graph datasets.

## 8 REPRODUCIBILITY STATEMENT

We provide code with the implementation of CLATT and instructions to reproduce all experimental results in our paper in the following anonymous repository: https://anonymous.4open.science/r/cluster_attention-F2BF.

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

# A DATASETS

## A.1 DATA SPLITS

In our experiments, we aim to use a range of datasets from different real-world applications of graph machine learning. Brief descriptions of these datasets are provided in Section 5.1. For datasets from the GraphLand benchmark (Bazhenov et al., 2025) (`city-roads-M`, `city-roads-L`, `avazu-ctr`, `tolokers-2`, `hm-categories`, `pokec-regions`) we use the official RL (random low) data splits, which are random stratified 10%/10%/80% train/val/test data splits. Similarly, for the `lastfm-asia` (Rozemberczki & Sarkar, 2020) and `facebook` (Rozemberczki et al., 2019) datasets, as well as for our new `amazon-ratings-full` dataset, we use random stratified 10%/10%/80% train/val/test data splits (see Appendix C for more details on `amazon-ratings-full` splits). The datasets `amazon-ratings-5core` and `questions` from Platonov et al. (2023b) originally come with 10 official random stratified 50%/25%/25% train/val/test data splits. However, we notice that when, as in our work, extensive hyperparameter tuning on the validation set is used (which is very important to achieve the best GNN performance in practice), using 10 different train/val/test splits leads to a certain type of test information leakage, as nodes that appear in the test data subset in one split can appear in val subsets in other splits. Thus, instead of using 10 data splits, we use only one data split in our experiments. Specifically, we use the first data split from the official 10 data splits.

## A.2 DATASET CHARACTERISTICS

The datasets used in our experiments are diverse in domains, sizes, and graph structural characteristics. Some structural characteristics of the considered graphs are provided in Table 1. First, notice that the datasets range in size from 7.6K nodes to 1.6M nodes and from 27.8K edges to 22.3M edges. We use both rather sparse and rather dense datasets, with average node degree ranging from 3.26 to 460.92 and median node degree ranging from 1 to 71. The average distance between two nodes in the considered graphs ranges from 2.45 to 194.05, while the maximum distance (graph diameter) ranges from 11 to 553. Clustering coefficients show how typical it is for two neighbors of a node to also be neighbors. There are two widely used definitions of clustering coefficients (Boccaletti et al., 2014): the global clustering coefficient and the average local clustering coefficient. In graphs used in our experiments, both clustering coefficients range from approximately zero to rather high values. Note that high values of clustering coefficients are not required for the strong performance of CLATT, as it can bring substantial benefits even on datasets with approximately zero clustering coefficients such as `city-roads-M`, `city-roads-L`, and `questions`. The degree assortativity coefficient is the Pearson correlation coefficient of degrees for pairs of linked nodes. Among the considered datasets, there are graphs with both positive, approximately zero, and negative degree assortativity coefficients.

Let us also discuss the relationships between graph structure and node labels in the considered datasets. Datasets where nodes tend to connect to other nodes with similar labels are known as *homophilous*, whereas datasets which do not exhibit this connectivity pattern are known as `non-homophilous`. To measure the levels of homophily for regression datasets, we use the target assortativity coefficient, which is the Pearson correlation coefficient of targets for pairs of linked nodes. To measure the levels of homophily for classification datasets, we use unbiased homophily (Mironov & Prokhorenkova, 2024) (with the $\alpha$ parameter set to 0), which satisfies more properties desirable for a homophily measure from Platonov et al. (2023a) than other commonly used homophily measures. Among the datasets used in our experiments, there is a range of both homophilous datasets (`city-roads-M`, `city-roads-L`, `pokec-regions`, `lastfm-asia`, `facebook`) and non-homophilous datasets (`avazu-ctr`, `tolokers-2`, `hm-categories`, `questions`, `amazon-ratings-5core`, `amazon-ratings-full`). As our experimental results show, CLATT can substantially improve performance on both of these types of datasets.

Note that some of the graphs considered in our experiments are directed, however, we convert all graphs to undirected ones both for experiments and for reporting statistics in Table 1.

## B  EXPERIMENTAL SETUP

For our experiments, we follow the protocol from Bazhenov et al. (2025). To compute the mean and standard deviation of model results, we train each model 10 times with different random seeds, except for the largest considered dataset `pokec-regions`, for which we train each model 5 times.

We train all our models in a full-batch setting as is common for GNNs, i.e., we do not use any subgraph sampling methods and train the models on the full graph (in particular, for GraphSAGE we only use the model architecture but not the neighbor sampling technique).

Hyperparameter tuning is extremely important for achieving optimal performance with GNNs. Thus, we conduct extensive hyperparameter grid search on the validation set for all models. Specifically, for learning rate we consider the values $\{3 \times 10^{-5}, 1 \times 10^{-4}, 3 \times 10^{-4}, 1 \times 10^{-3}, 3 \times 10^{-3}\}$, and for dropout we consider the values of $\{0, 0.1, 0.2\}$. For datasets from the GraphLand benchmark, we additionally consider transforming numerical features with either standard scaling or quantile transformation to standard normal distribution, and we for regression datasets, we additionally consider either transforming targets with standard scaling or leaving them untransformed. For all models, we set the hidden dimension to 512. For GGTs, we set the dimension of positional encodings (DeepWalk embeddings or Laplacian eigenvectors) to 128. We train all models for 1000 steps with Adam optimizer (Kingma & Ba, 2015), except for the `amazon-ratings-5core` and `amazon-ratings-full` datasets, for which we found longer training to be beneficial, and thus train all models for 3000 steps.

Our model implementations use PyTorch (Paszke et al., 2019) and DGL (Wang et al., 2019).

## C  AMAZON-RATINGS-FULL DATASET

The `amazon-ratings-5core` dataset from Platonov et al. (2023b) (originally called `amazon-ratings`) was obtained from data collected by Leskovec et al. (2007). Since this datasets was used by Platonov et al. (2023b) to evaluate models specifically designed for non-homophilous graphs, many of which are not scalable, only the 5-core of the graph (Malliaros et al., 2020) was used, i.e., nodes of degree less than 5 were iteratively removed from the graph until no such nodes were left. While this procedure was used to reduce the size of the graph, we notice that it resulted in a graph with a peculiar structure: `amazon-ratings-5core` has a lot of small clusters of 5 or slightly more nodes that are very densely interconnected (often being cliques or almost cliques) but connected with the rest of the graph with only one or two edges. We hypothesize that such graph structure might be particularly amenable to node clustering and might overestimate the performance of CLATT (compared to its performance on the full co-purchasing network). Thus, we constructed a new dataset — `amazon-ratings-full` — which follows the same dataset construction process as used in Platonov et al. (2023b) but skips the reduction of the graph to its 5-core. Thus, we obtain the full Amazon co-purchasing network from Leskovec et al. (2007) (more specifically, its full largest connected component, as the original data also contains many small connected components and isolated nodes) with the same task and node features as used by Platonov et al. (2023b). As can be seen from Table 1, `amazon-ratings-full` is more than an order of magnitude larger than `amazon-ratings-5core`, while having almost the same diameter but smaller average node degree, average distance between nodes, and unbiased homophily.

When creating data splits for our new dataset, we follow the approach from Bazhenov et al. (2025) and create 3 different data splits that can be used for different purposes. Specifically, we create 2 random stratified data splits with different proportions: the RL (random low) data split is a 10%/10%/80% train/val/test data split, while the RH (random high) data split is a 50%/25%/25% train/val/test data split. Further, the TH (temporal high) data split is a temporal data split with exactly the same proportions as the RH split, which can be used to evaluate model performance under the challenging setting of temporal distributional shifts. Since the original data does not have information about the time a product appeared on Amazon, we use the time the first review for a product appeared as a proxy for it to create the temporal data split. Additionally, the THI (temporal high / inductive) setting can be used with the `amazon-ratings-full` dataset by considering three different snapshots of the graph for training, validation, and testing. In all our experiments in this work, we use the RL (random low) data split for the `amazon-ratings-full` dataset, similar to our setup for datasets from the GraphLand benchmark.

Table 3: Distributions of per-node average attention distances. Note that the unweighted average pairwise distance is 5.23 for `lastfm-asia` and 2.79 for `tolokers-2`.

| | lastfm-asia | | | | | tolokers-2 | | | | |
|---|---|---|---|---|---|---|---|---|---|---|
| *quantile* | 0.05 | 0.25 | 0.50 | 0.75 | 0.95 | 0.05 | 0.25 | 0.50 | 0.75 | 0.95 |
| LGT (local att.) | 0.39 | 0.62 | 0.79 | 0.90 | 0.97 | 0.59 | 0.91 | 0.97 | 0.99 | 1.00 |
| LGT-CLATT (local att.) | 0.39 | 0.62 | 0.79 | 0.90 | 0.97 | 0.56 | 0.91 | 0.98 | 0.99 | 1.00 |
| LGT-CLATT (cluster att.) | 1.73 | 2.46 | 2.96 | 3.57 | 4.82 | 1.23 | 1.73 | 1.99 | 2.32 | 3.14 |
| GGT-DW (global att.) | 4.13 | 4.71 | 5.16 | 5.67 | 6.61 | 2.23 | 2.49 | 2.71 | 2.93 | 3.35 |
| GGT-DW-CLATT (global att.) | 4.11 | 4.87 | 5.35 | 5.91 | 6.87 | 2.26 | 2.52 | 2.73 | 2.94 | 3.36 |
| GGT-DW-CLATT (cluster att.) | 1.36 | 2.41 | 3.01 | 3.88 | 5.47 | 1.23 | 1.73 | 2.00 | 2.33 | 3.16 |
| GGT-Lap (global att.) | 4.18 | 4.71 | 5.13 | 5.65 | 6.59 | 1.96 | 2.15 | 2.82 | 3.00 | 3.90 |
| GGT-Lap-CLATT (global att.) | 3.88 | 4.62 | 5.07 | 5.60 | 6.55 | 2.24 | 2.50 | 2.71 | 2.93 | 3.34 |
| GGT-Lap-CLATT (cluster att.) | 1.68 | 2.43 | 2.92 | 3.52 | 4.73 | 1.20 | 1.72 | 1.99 | 2.32 | 3.14 |

We see that the improvements from CLATT are smaller on `amazon-ratings-full` than on `amazon-ratings-5core`, confirming that the specific graph structure of `amazon-ratings-5core` is particularly beneficial for CLATT. However, even on `amazon-ratings-full`, CLATT brings statistically significant increases in model performance for all the considered models.

## D  ATTENTION DISTANCES ANALYSIS

In this section, we investigate how far nodes typically look with different types of attention. Specifically, we analyze the average distances weighted with attention probabilities for the three considered graph attention mechanisms: local (1-hop neighborhood), global (full-graph), and cluster attention. We consider three base models — LGT (local attention), GGT-DW (global attention), GGT-Lap (global-attention), and their cluster attention augmented variants — LGT-CLATT, GGT-DW-CLATT, GGT-Lap-CLATT, and analyzae their attention patterns on two datasets — `lastfm-asia` and `tolokers-2`. We limit ourselves to these two datasets because they are the smallest among the ones considered in our work, and our analysis requires the full attention matrices for global and cluster attention, which take up a lot of memory (in our experiments, we use efficient FlashAttention-based (Dao et al., 2022) attention algorithms which do not materialize the full attention matrices and thus allow us to scale to even very large graphs, however, for the analysis in this section, we need the full attention matrices).

For each dataset, each model, and each attention head, we compute the average distance to the nodes to which each node attends weighted by the corresponding attention probabilities, thus obtaining $nk$ distances per dataset and model, where $n$ is the number of nodes in the graph and $k$ is the number of attention heads in the model. In Table 3, we report the 0.05, 0.25, 0.50, 0.75, 0.95 quantiles of these $nk$ distances, thus summarizing their distributions.

First, let us note that the average attention distances for cluster and global attention are significantly different between the `lastfm-asia` and `tolokers-2` datasets, which is explained by their significantly different graph structure: in `lastfm-asia`, the unweighted average pairwise distance between nodes is a lot larger than in `tolokers-2`, and thus attention also typically works at greater distances in `lastfm-asia` than in `tolokers-2`. Also note that for local attention, the average attention distance is upper bounded by 1, since nodes can only attend to themselves (at a distance of 0) and to their direct neighbors (at a distance of 1). We can see that in `lastfm-asia` nodes attend to themselves in local attention significantly more than in `tolokers-2`, which is likely due to `lastfm-asia` being an a lot more homophilous dataset.

Now, let us focus on comparing cluster attention to other types of attention. We can see that in cluster attention, nodes on average attend at significantly longer distances than in local attention (in which the maximum distance is limited to 1), and often even at longer distances than the maximum receptive field of a multilayer local-attention GNN (which for this experiment is 3 since 3-layer GNNs are used). Specifically, in `lastfm-asia`, more than 25% of nodes attend at distances larger than 3.5 on average, while in `tolokers-2` more than 5% of nodes attend at distances larger than 3.1 on average, which are distances unattainable for local attention even with 3 layers. However,

in cluster attention nodes also on average attend at significantly smaller distances than in global attention. Specifically, in both `lastfm-asia` and `tolokers-2`, more than half of the nodes attend on average at a distance smaller than even the $0.05$ quantile of nodes in global attention (and this difference in distances is more than 1 hop for `lastfm-asia`).

Our results from Section 5 show that adding cluster attention consistently improves the performance of both local-attention and global-attention models, and our observations from this section provide an explanation for why this happens. It can be useful for a model to pass information beyond the receptive field of local-attention models (and other MPNNs), which can be achieved with cluster attention. While global-attention models also provide an opportunity for using long-range interactions, due to their lack of graph-based inductive biases, they struggle to focus on nearby nodes and often use attention to attend further than is actually needed, which also can be mitigated with cluster attention.

