# OpenReview forum: "Cluster Attention for Graph Machine Learning"
_ICLR.cc/2026/Conference — Submitted to ICLR 2026_

### Official Review · Reviewer_Cdbo · 2025-10-24

**Soundness:** 3
**Presentation:** 2
**Contribution:** 2
**Rating:** 2
**Confidence:** 4

**Summary:**

This paper proposes a mechanism named Cluster Attention (CLATT), which applies attention over graph clustering results to bridge the gap between traditional message passing and global attention. However, the work has room for improvement in terms of theoretical depth, novelty margin, and comprehensiveness of baseline comparisons.

**Strengths:**

1. The motivation is reasonable; using graph clustering algorithms as a preprocessing step and applying attention is an intuitive and rational intermediate solution.
2. The experimental datasets cover a wide range.

**Weaknesses:**

1. Lack of novelty. Information aggregation over clusters/communities is not a new concept in graph learning [1]. The paper does not discuss its connections to or distinctions from related work such as Graph Coarsening, hierarchical GNNs, or hypergraph studies.
2. Lack of theoretical analysis. Although the method design is intuitive, it would be beneficial to include a section (even informal one) discussing the underlying mechanisms that explain why CLATT works.
3. Insufficient experiments. The evaluation lacks analysis of computational overhead and the impact of different clustering algorithms. Additionally, several relevant baselines are missing, such as the classic GAT and recent state-of-the-art methods.
4. Some parts of the writing are overly informal (e.g., lines 186–203), and the overall presentation lacks clarity and structure.

[1] Bianchi F M, Grattarola D, Alippi C. Spectral clustering with graph neural networks for graph pooling[C]//International conference on machine learning. PMLR, 2020: 874-883.

**Questions:**

See the weaknesses mentioned above.

**Details Of Ethics Concerns:**

No.

---

> ### Author Response · Authors · 2025-11-18
> **Author Rebuttal Part 1/2**
>
> Thank you for your feedback. We address your concerns below.
>
> > Lack of novelty. Information aggregation over clusters/communities is not a new concept in graph learning [1]. The paper does not discuss its connections to or distinctions from related work such as Graph Coarsening, hierarchical GNNs, or hypergraph studies.
>
> Indeed, there are prior works bridging graph clustering and GNNs, however, they typically focus on using GNNs to cluster a graph (also referred to as graph pooling) often optimizing some version of soft modularity, while our work is to the best of our knowledge the first one to propose using graph clusterings computed with classic algorithms to create a new type of structure-aware graph attention and use it for standard node property prediction tasks.
>
> Regarding more specifically the work [1] mentioned by you: this work focuses on graph clustering with GNNs. While among its experiments it provides results for classic node classification datasets cora, citeseer, and pubmed, it does not actually perform semi-supervised node classification on them, but rather performs unsupervised node clustering and then matches the obtained clusters to ground-truth classes. In contrast, our method can be directly applied to any node property prediction task.
>
> > Lack of theoretical analysis. Although the method design is intuitive, it would be beneficial to include a section (even informal one) discussing the underlying mechanisms that explain why CLATT works.
>
> Formally analyzing CLATT is highly non-trivial, as it combines graph clustering with attention, which introduces potential complicated interaction between the clustering algorithm and the GNN. In particular, CLATT can use any graph clustering, which provides a very significant degree of freedom, as different graph clustering algorithms rely on different assumptions and have widely different theoretical properties. However, we do not consider the lack of theoretical guarantees to be a significant limitation of our work, as our main aim is to introduce a new attention variant for graph data and demonstrate its practical usefulness. Similarly, for other graph ML models, such as GNNs and Graph Transformers, the theoretical analysis was developed much later than the models were introduced. However, in the updated version of our paper, we have added Appendix D in which we analyze attention patterns of CLATT and other models, which provides insight into why CLATT is useful. On the on hand, we show that cluster attention typically works at a significantly longer range than local 1-hop interactions of classic MPNNs, and often even at a longer range than the entire receptive field of multilayer MPNNs, showing that longer-range interactions than allowed in MPNNs are useful. On the other hand, we show that cluster attention typically works at a smaller range than Global Graph Transformers, which, combined with CLATT’s empirical performance, suggests that there are useful mid-range interactions on which Global Graph Transformers fail to focus enough due to their lack of structure-based inductive biases.

---

> > ### Author Response · Authors · 2025-11-18
> > **Author Rebuttal Part 2/2**
> >
> > > Insufficient experiments. The evaluation lacks analysis of computational overhead and the impact of different clustering algorithms. Additionally, several relevant baselines are missing, such as the classic GAT and recent state-of-the-art methods.
> >
> > Regarding state-of-the-art methods: please note that we use a selection of classic MPNNs and Global Graph Transformers. We use their implementations with architectural improvements such as skip-connections, layer normalization, and additional MLP blocks, which have been shown to significantly improve the performance of classic GNNs and make them current state-of-the-art methods [2-5].
> >
> > As for GAT, it is indeed a classic method that we missed in our experiments, and we provide its results in the table below:
> >
> > |                      | GAT          | GAT-CLATT    |
> > | -------------------- | ------------ | ------------ |
> > | city-roads-M         | 59.11 ± 0.20 | 60.31 ± 0.24 |
> > | city-roads-L         | 53.43 ± 0.20 | 55.38 ± 0.24 |
> > | avazu-ctr            | 33.20 ± 0.20 | 33.54 ± 0.26 |
> > | tolokers-2           | 53.41 ± 1.32 | 55.29 ± 0.56 |
> > | hm-categories        | 68.09 ± 0.35 | 68.91 ± 0.50 |
> > | pokec-regions        | 45.87 ± 0.32 | 49.84 ± 0.40 |
> > | lastfm-asia          | 83.04 ± 0.27 | 84.12 ± 0.45 |
> > | facebook             | 92.54 ± 0.19 | 92.85 ± 0.28 |
> > | questions            | 19.04 ± 0.57 | 21.27 ± 0.84 |
> > | amazon-ratings-5core | 51.39 ± 0.54 | 54.35 ± 0.31 |
> > | amazon-ratings-full  | 38.75 ± 0.13 | 39.02 ± 0.13 |
> >
> > It can be seen that CLATT consistently improves performance for GAT, just as for the other models.
> >
> > As for computational overhead: CLATT does not lead to significant time overhead as it can be implemented using efficient FlashAttenmtion-based dense attention algorithms. As an example, we provide running times for our largest considered dataset which is pokec-regions with more than 1.5 million nodes and more than 22 million edges (note that this is orders of magnitude larger than most popular datasets in modern graph machine learning such as cora and citeseer). For this dataset, our heaviest MPNN - LGT - takes 25 minutes for a single run, while LGT-CLATT takes 31 minutes.
> >
> > > Some parts of the writing are overly informal (e.g., lines 186–203), and the overall presentation lacks clarity and structure.
> >
> > We will improve our presentation and make the specified part more formal. If you have any examples of other parts that lack clarity or structure, we would be grateful if you could specify them.
> >
> > [2] A critical look at the evaluation of GNNs under heterophily: Are we really making progress? (ICLR 2023)
> >
> > [3] Classic GNNs are Strong Baselines: Reassessing GNNs for Node Classification (NeurIPS 2024)
> >
> > [4] Can Classic GNNs Be Strong Baselines for Graph-level Tasks? Simple Architectures Meet Excellence (ICML 2025)
> >
> > [5] GraphLand: Evaluating Graph Machine Learning Models on Diverse Industrial Data (NeurIPS 2025)

---

### Official Review · Reviewer_zGQs · 2025-10-28

**Soundness:** 2
**Presentation:** 2
**Contribution:** 2
**Rating:** 4
**Confidence:** 3

**Summary:**

This paper introduces CLATT, a graph neural network component that allows nodes to attend to all other nodes within the same cluster, as determined by community detection or graph clustering algorithms. The key motivation is to provide an inductive bias that leverages graph structure for expanding receptive fields, striking a balance between pure message passing, which is limited to local neighborhoods, and global transformers, which ignore graph structure. The authors show how CLATT can be integrated into both MPNNs and global graph transformers, then empirically evaluate the approach on 11 diverse datasets, demonstrating generally improved performance.

**Strengths:**

1. Proposes a clear and well-motivated approach that uses graph clustering to define mid-range attention, addressing the limitations of both local message passing and indiscriminate global attention. This structural bias is conceptually appealing for tasks where communities/clusters are meaningful.

2. Experiments cover a wide range of datasets including both homophilous and non-homophilous graphs, sparse and dense graphs, and several realistic/industrial datasets, not just citation benchmarks.

3. The results in Table 2 demonstrate consistent improvements across several strong baselines, including GCN, GraphSAGE, Local Graph Transformers, and two types of global graph transformers with various positional encodings. For example, performance improvements for GCN-CLATT and GraphSAGE-CLATT are evident and statistically significant in most cases.

**Weaknesses:**

1. While the paper goes into some mathematical detail on the CLATT mechanism, there is almost no theoretical analysis of its representational or generalization properties. For instance, there is no discussion of what information might be lost or gained under various clustering regimes, no bounds on expressivity, and no framework for reasoning about which graph types CLATT is provably suitable for.

2. The concatenation of multiple cluster-based attention outputs for each node may cause the embedding size to scale linearly with the number of clusterings. This could introduce issues with memory and computation for large graphs or when many clustering algorithms are used.

3. While the key equations for CLATT are present (Eq., Pages 4-5), their notation is sometimes ambiguous. For example, there’s no clarification about the treatment of singleton clusters, solution for clusters of widely varying size, or masking strategy when clusters overlap (if allowed). Additionally, the use of learnable parameters per clustering ($\mathbf{W}_{q}^{C}$, etc.) has unclear implications for parameter sharing and scalability—none of these architectural choices are ablated/justified.

**Questions:**

1. How robust is CLATT to cluster quality? Have the authors tried intentionally poor clustering (e.g., random partitions) to probe sensitivity?
Does the concatenation-based combination of multiple clusterings risk dimensionality inflation or feature redundancy, and are there alternatives (e.g., learned aggregation) that trade off better between expressivity and complexity?

2. Are there meaningful differences in which types of graphs (homophilous, highly modular, assortative/disassortative) benefit most from CLATT, and can the authors provide a diagnostic or criterion to recommend when CLATT should or shouldn’t be applied?

3. Can the authors provide more insight or empirical results for tasks where CLATT fails or offers no benefit?

---

> ### Author Response · Authors · 2025-11-18
> **Author Rebuttal Part 1/2**
>
> Thank you for your review! We address your concerns below.
>
> > While the paper goes into some mathematical detail on the CLATT mechanism, there is almost no theoretical analysis of its representational or generalization properties. For instance, there is no discussion of what information might be lost or gained under various clustering regimes, no bounds on expressivity, and no framework for reasoning about which graph types CLATT is provably suitable for.
>
> Formally analyzing CLATT is highly non-trivial, as it combines graph clustering with attention, which introduces potential complicated interaction between the clustering algorithm and the GNN. In particular, CLATT can use any graph clustering, which provides a very significant degree of freedom, as different graph clustering algorithms rely on different assumptions and have widely different theoretical properties. However, we do not consider a lack of theoretical guarantees to be a significant limitation of our work, as our main aim is to introduce a new attention variant for graph data and demonstrate its practical usefulness. Similarly, for other graph ML models, such as GNNs and Graph Transformers, the theoretical analysis was developed much later than the models were introduced. However, in the updated version of our paper, we have added Appendix D in which we analyze attention patterns of CLATT and other models, which provides insight into why CLATT is useful. On the one hand, we show that cluster attention typically works at a significantly longer range than local 1-hop interactions of classic MPNNs, and often even at a longer range than the entire receptive field of multilayer MPNNs, showing that longer-range interactions than allowed in MPNNs are useful. On the other hand, we show that cluster attention typically works at a smaller range than Global Graph Transformers, which, combined with CLATT’s empirical performance, suggests that there are useful mid-range interactions on which Global Graph Transformers fail to focus enough due to their lack of structure-based inductive biases.
>
> > The concatenation of multiple cluster-based attention outputs for each node may cause the embedding size to scale linearly with the number of clusterings. This could introduce issues with memory and computation for large graphs or when many clustering algorithms are used.
>
> Indeed, using many different clusterings leads to an increase in dimension, which is why we limit ourselves to 4 clusterings at most, and for most datasets we only use 1 or 2 clusterings. If working with more clusterings is required for some reason, node representations can be projected to a smaller dimension. We would like to note that all our experiments fit into a standard 80GB GPU, even for our largest pokec-regions dataset, which has more than 1.5 million nodes and more than 22 million edges and is orders of magnitude larger than datasets commonly used in modern graph machine learning such as cora and citeseer.
>
> > While the key equations for CLATT are present (Eq., Pages 4-5), their notation is sometimes ambiguous. For example, there’s no clarification about the treatment of singleton clusters, solution for clusters of widely varying size, or masking strategy when clusters overlap (if allowed). Additionally, the use of learnable parameters per clustering ($\mathbf{W}_{q}^{C}$, etc.) has unclear implications for parameter sharing and scalability—none of these architectural choices are ablated/justified.
>
> We discard singleton clusters in CLATT, i.e., these nodes do not attend to anything. There can be no cluster overlap in all the considered clustering algorithms. To work with clusters of different size, either padding can be applied to make all sizes the same (with appropriate attention masking) or ragged/jagged tensors provided by modern DL frameworks like PyTorch and TensorFlow can be used which allow stacking different-length sequences into a tensor-like structure and efficiently working with them without padding. We will clarify these details in the text. Learnable parameters per clustering are necessary as different clusterings can have very different underlying meanings, however, it does not lead to a significant increase in parameter counts as we limit the number of clusterings used (see above), and in Graph Transformer models most parameters are in MLP blocks rather than in attention blocks anyway.

---

> ### Author Response · Authors · 2025-11-18
> **Author Rebuttal Part 2/2**
>
> > How robust is CLATT to cluster quality? Have the authors tried intentionally poor clustering (e.g., random partitions) to probe sensitivity? Does the concatenation-based combination of multiple clusterings risk dimensionality inflation or feature redundancy, and are there alternatives (e.g., learned aggregation) that trade off better between expressivity and complexity?
>
> CLATT relies on meaningful clusters. With meaningless/random clusters, models with CLATT perform on the level of models without CLATT. To counter dimensionality increase with multiple clusterings, projections to lower dimensions can be used, however, in practice, we did not run into memory or running time issues with CLATT.
>
> > Are there meaningful differences in which types of graphs (homophilous, highly modular, assortative/disassortative) benefit most from CLATT, and can the authors provide a diagnostic or criterion to recommend when CLATT should or shouldn’t be applied?
>
> As our experiments show, CLATT works on a wide variety of graphs with different structural properties, both homophilous and heterophilous, which suggests that CLATT is a rather general method. We expect that CLATT will not work only on graphs with lack of meaningful cluster structure, such highly regular grid-like graphs. However, as classic results from network science show, most real-world social, information, and biological networks exhibit well-defined clusters.
>
> > Can the authors provide more insight or empirical results for tasks where CLATT fails or offers no benefit?
>
> As mentioned above, CLATT does not provide benefits on graphs with no meaningful graph structure. For example, CLATT does not increase performance on the minesweeper dataset from [1], which is a semi-synthetic dataset representing the classic minesweeper game with a 2D grid graph.
>
>
>
> [1] A critical look at the evaluation of GNNs under heterophily: Are we really making progress? (ICLR 2023)

---

### Official Review · Reviewer_cWK1 · 2025-10-28

**Soundness:** 3
**Presentation:** 2
**Contribution:** 2
**Rating:** 4
**Confidence:** 5

**Summary:**

The paper introduces Cluster Attention (CLATT), an attention layer for graphs where each node attends over nodes grouped by precomputed clusters rather than only 1-hop neighbors (MPNNs) or the entire graph (global transformers). The authors instantiate CLATT with four candidate clustering methods. They select a small subset of clusterings that are mutually dissimilar, apply CLATT separately to each selected clustering, and concatenate the outputs. They report gains across diverse benchmarks (including GraphLand) when inserting CLATT into GCN/GraphSAGE and local/global transformers.

**Strengths:**

The paper identifies a pragmatic middle ground between local and global aggregation; it offers a general plug-in, reports results on many datasets, and discusses engineering choices (handling different clusterings, concatenation and projection). The candidate set of clusterers covers assortative, disassortative, and feature-space partitions, which is a sensible coverage of regimes encountered in practice.

**Weaknesses:**

1. Although all four candidate algorithms are present in the paper, they are not enumerated in one place with assumptions and hyperparameters. Please add a compact summary of them.

2. The paper states a desire to pick diverse clusterings. You do show similarity matrices and mention the Correlation Coefficient (CC) as the similarity index; however, the selection rule (how CC informs which clusterings are kept) is not formalized. Please specify:

* the metric used,
* the threshold or diversity objective (e.g., greedy selection maximizing minimum pairwise dissimilarity),
* whether similarity is computed on the full graph or a validation subset, and its computational cost.

3. Combining clustering with GNNs is not a novel idea [1,2]. Position CLATT more directly against:

- GNN-based clustering and end-to-end formulations [1];
- Hierarchical / coarsened MPNNs and hierarchical message passing [2];
- global transformers with structural encodings (e.g., Laplacian/positional encodings, shortest-path/centrality biases).
  Clarify where CLATT provides a new capability (e.g., multi-clustering selective aggregation) rather than a re-packaging of hierarchical pooling.

4. “Is attention necessary?” Because CLATT attends within clusters, a simple baseline is to pool each cluster (mean/max/gated pooling) and pass pooled summaries to nodes (or to use cluster-level tokens with cross-attention). Without this, it is hard to attribute gains to attention rather than to the cluster prior itself. [2]

5. The paper’s motivation is that global attention lacks graph-topology bias. CLATT’s topology-based clusterers do inject such bias, but the inclusion of feature-only k-means weakens the story. Please either:
(i) frame CLATT as a general prior layer that can combine topology and feature clusterings, and analyze when each helps; or
(ii) restrict main claims to topology-derived clusterings and move feature-only variants to ablations.

6. Compute and memory. Report wall-clock and peak memory for base vs. CLATT (including clustering and selection overhead) on the largest datasets. This is important for practitioners.

**Questions:**

1. Please confirm the four candidate methods (Leiden/CPM; Bayesian planted partition; disassortative hierarchical SBM with smallest non-trivial level; k-means on ResMLP embeddings) and provide their hyperparameters and implementations used.

2. How exactly do you quantify differences between clusterings? How are diversity thresholds chosen, and how does this choice affect accuracy and cost?

3. Why not learn clusters with a GNN-based clustering module [1], potentially allowing backpropagation of the task loss into the clustering? Could CLATT benefit from such end-to-end training?

4. Is attention necessary? Please add ablations replacing CLATT with cluster pooling (mean/max/gated) or cluster tokens without attention, to show the specific value of attention beyond the cluster prior. See [2] for hierarchical message passing baselines.

5. Can you provide analyses showing that CLATT improves structure-sensitive metrics (e.g., performance vs. assortativity, conductance, role similarity) when using topology-only clusterers, and contrast that with feature-only k-means?

6. What is the delta when re-selecting the clusterings per backbone?

7. How sensitive is CLATT to the resolution (average cluster size) across the topology-based methods?

[1] Graph clustering with graph neural networks, JMLR, 2023.

[2] Hierarchical message-passing graph neural networks, DMKD, 2023.

---

> ### Author Response · Authors · 2025-11-18
> **Author Rebuttal Part 1/3**
>
> Thank you for your review! We reply to your questions below.
>
> > Please confirm the four candidate methods (Leiden/CPM; Bayesian planted partition; disassortative hierarchical SBM with smallest non-trivial level; k-means on ResMLP embeddings) and provide their hyperparameters and implementations used.
>
> For the Leiden algorithm, Bayesian Planted Partition, and disassortative SBM, we use the official implementations from the corresponding papers with default hyperparameters. Specifically, for the Leiden algorithm we use the leidenalg library (https://leidenalg.readthedocs.io/), and for Bayesian Planted Partition and disassortative SBM we use the graph-tool library (https://graph-tool.skewed.de/static/docs/stable/autosummary/graph_tool.inference.PPBlockState.html#graph_tool.inference.PPBlockState, https://graph-tool.skewed.de/static/docs/stable/autosummary/graph_tool.inference.minimize_nested_blockmodel_dl.html#graph_tool.inference.minimize_nested_blockmodel_dl). For k-means, we use the implementation from sklearn (https://scikit-learn.org/stable/modules/generated/sklearn.cluster.KMeans.html) with 150 clusters and default values for other hyperparameters (the other algorithms do not need a predetermined number of clusters). We will add an appendix with this information.
>
> > How exactly do you quantify differences between clusterings? How are diversity thresholds chosen, and how does this choice affect accuracy and cost?
>
> We would like to clarify that we do not use cluster diversity metrics to select clusterings, we only provide them to confirm that different algorithms indeed find significantly different clusterings. The procedure for the choice of the clustering algorithms for different datasets is described in lines 273-279. Specifically, we treat the choice of clustering algorithms as a hyperparameter, run an experiment with each of the four clusterings, and select those clusterings that demonstrated a performance increase in these runs. Note that for all models and datasets, we run an extensive hyperparameter search (described in Appendix B) with several dozens of hyperparameter values, and thus the cost of adding four more runs for choosing clusterings is negligible.
>
> > Combining clustering with GNNs is not a novel idea [1,2]. Position CLATT more directly against:
> GNN-based clustering and end-to-end formulations [1];
> Hierarchical / coarsened MPNNs and hierarchical message passing [2];
> global transformers with structural encodings (e.g., Laplacian/positional encodings, shortest-path/centrality biases). Clarify where CLATT provides a new capability (e.g., multi-clustering selective aggregation) rather than a re-packaging of hierarchical pooling.
>
> Indeed, there are prior works bridging graph clustering and GNNs, however, they typically focus on using GNNs to cluster a graph (also referred to as graph pooling) often optimizing some version of soft modularity, while our work is to the best of our knowledge the first one to propose using graph clusterings computed with classic algorithms to create a new type of structure-aware graph attention and use it for standard node property prediction tasks.
>
> Regarding more specifically the works mentioned by you:
>
> [1] is a classic example of using GNNs to cluster a graph, which we cite in the Related Work section as an example of graph clustering algorithms.
>
> [2] enhances message-passing by adding hierarchical super-nodes to the graph. In contrast to our work, it uses neither classic graph clustering algorithms nor attention. While adding virtual nodes representing graph clusters is a possible approach to utilize cluster information, in our preliminary experiments, we found it to be significantly inferior to cluster attention, since it introduces a bottleneck for all within-cluster interactions, while in cluster attention any node can directly attend to any other node within each cluster without any limits to information exchange.
>
> Regarding Global Graph Transformers with structural encoding, they also do not explicitly use any cluster information. As our experiments show, adding cluster attention to Global Graph Transformers significantly improves their performance, which suggests that without it GGTs do not have enough structural biases. We have added Appendix D to the updated version of our paper, in which we demonstrate that global attention typically operates at much longer distances in the graph than cluster attention, which suggests that GGTs struggle to focus on nearby nodes and attend at longer ranges than is actually useful without more explicit structural information provided.

---

> > ### Author Response · Authors · 2025-11-18
> > **Author Rebuttal Part 2/3**
> >
> > > Is attention necessary? Please add ablations replacing CLATT with cluster pooling (mean/max/gated) or cluster tokens without attention, to show the specific value of attention beyond the cluster prior. See [2] for hierarchical message passing baselines.
> >
> > As we have mentioned above, we ran preliminary experiments in which instead of cluster attention we used per-cluster virtual nodes or pooling of nodes within the same cluster, and it worked significantly worse than cluster attention. This is likely due to the fact that these methods try to pack all cluster information into a single vector, thus introducing a bottleneck, while cluster attention allows any node to directly exchange information with any other node within the same cluster. We provide the results of these experiments below.
> >
> > |                  | city-roads-M | tolokers-2   | hm-categories | amazon-ratings-5core |
> > | ---------------- | ------------ | ------------ | ------------- | -------------------- |
> > | LGT              | 58.05 ± 0.58 | 55.70 ± 0.28 | 69.25 ± 0.25  | 51.26 ± 0.38         |
> > | LGT + virt. node | 58.41 ± 0.32 | 55.94 ± 0.21 | 69.12 ± 0.37  | 51.93 ± 0.29         |
> > | LGT + pooling    | 58.78 ± 0.45 | 56.12 ± 0.38 | 69.45 ± 0.40  | 52.21 ± 0.31         |
> > | LGT-CLATT        | 60.05 ± 0.31 | 56.75 ± 0.34 | 70.25 ± 0.35  | 53.75 ± 0.36         |
> >
> > > Why not learn clusters with a GNN-based clustering module [1], potentially allowing backpropagation of the task loss into the clustering? Could CLATT benefit from such end-to-end training?
> >
> > Combining cluster attention and cluster learning is a very complicated task. First, end-to-end graph clustering with GNNs produces soft clusterings, while CLATT operates over classic hard clusterings, and converting soft clusterings to hard ones will break differentiability. Second, when applied to a fixed clustering, as we do, CLATT can be significantly optimized by compiling the attention operation (which modern DL libraries like JAX and PyTorch allow), which leads to excellent scalability. If clustering will be learned end-to-end and thus evolve over the course of a single training run, such compilation will not be possible and performance will significantly degrade. Overall, while it can be an interesting direction for further research, we believe it is out of the scope of our current work.
> >
> > > the inclusion of feature-only k-means weakens the story. Please either: (i) frame CLATT as a general prior layer that can combine topology and feature clusterings, and analyze when each helps; or (ii) restrict main claims to topology-derived clusterings and move feature-only variants to ablations.
> >
> > We included the feature-based clustering to demonstrate that it is one more possible way of clustering graph nodes in attributed graphs, however, we agree that is a bit unrelated to our main approach. In fact, in our experiments, we only use feature-based clustering for a single dataset (questions), as for all the other ones it was not chosen by our hyperparameter selection procedure. We will improve our text to emphasize that we focus on structure-based clustering and move feature-based clustering to additional experiments in the Appendix.
> >
> > > Can you provide analyses showing that CLATT improves structure-sensitive metrics (e.g., performance vs. assortativity, conductance, role similarity) when using topology-only clusterers, and contrast that with feature-only k-means?
> >
> > Could you please clarify what you mean here? The graph remains unchanged during training and thus structural characteristics like assortativity also do not change. However, as we show in our experiments, CLATT significantly improves classification/regression metrics.
> >
> > > What is the delta when re-selecting the clusterings per backbone?
> >
> > While clustering algorithms are randomized, we find that the produced clusterings are highly similar and do not affect model performance beyond the reported standard deviations.
> >
> > > How sensitive is CLATT to the resolution (average cluster size) across the topology-based methods?
> >
> > In our experiments, we use graphs with widely different sizes, densities, and other structural characteristics, thus leading to clustering algorithms finding clusters of very different sizes for different graphs (and even within a single graph), however, as we demonstrate, CLATT works well for all these graphs.

---

> > > ### Author Response · Authors · 2025-11-18
> > > **Author Rebuttal Part 3/3**
> > >
> > > > Compute and memory. Report wall-clock and peak memory for base vs. CLATT (including clustering and selection overhead) on the largest datasets. This is important for practitioners.
> > >
> > > Our largest considered dataset is pokec-regions with more than 1.5 million nodes and more than 22 million edges (note that this is orders of magnitude larger than most popular datasets in modern graph machine learning such as cora and citeseer). For this dataset, our heaviest MPNN - LGT - takes 25 minutes for a single run, while LGT-CLATT takes 31 minutes. Global Graph Transformers are significantly less efficient, with GGT taking 40 hours and GGT-CLATT taking 45 hours (this is why we report TL for these methods). However, we can see that adding CLATT does not add much time overhead compared to the base model, whether it is a local MPNN or a Global Graph Transformer. We would also like to note that the main obstacle to scaling GNNs to large graphs is typically memory rather than time, and CLATT does not introduce significant memory overhead as efficient FlashAttention-based algorithms can be used that do not materialize the full attention matrix. All our experiments fit in a single 80GB GPU.

---

### Official Review · Reviewer_YsWn · 2025-10-31

**Soundness:** 3
**Presentation:** 3
**Contribution:** 3
**Rating:** 6
**Confidence:** 4

**Summary:**

The paper introduces Cluster Attention (CLATT): run one or more fast graph clustering/community-detection algorithms as a preprocessing step, then perform dense attention within each cluster (optionally for multiple clusterings and concatenate their outputs), and combine this with standard message passing (or with global graph transformers). The goal is a “middle ground” between purely local MPNNs and fully global attention—capturing longer-range dependencies while retaining graph-structural inductive biases.

**Strengths:**

1. This paper strikes a practical design point between edge-local MPNN and full all-pairs GGT -- computing graph attention on edges and nodes within each cluster.
2. Despite relying on one clustering algorithm, the proposed method considers various different clustering algorithms and combine the different types of in-cluster attention.
3. This paper covers diverse types of graph datasets to validate the effectiveness.

**Weaknesses:**

1. The complexity is $\sum |C_i|^2$ where $|C_i|$ is the number of cluster i. However, sometimes most of the nodes in a graph will be clustered into one cluster. For example, when the graph has a majority connected component and many small unconnected parts. In this condition, the computational complexity is $O(|V|^2)$ and the proposed method will be very similar to graph Transformer.
2. Other case where the proposed method will not work well include grids.
3. In inductive settings, unseen nodes during training may significantly change the clustering structure of graphs. And thus the learned attention may not be appropriate for the real underlying clustering structure.
4. There are already some previous works [1,2,3] combining graph clustering and graph attention, the authors should discuss the difference between their proposed method and previous literature.
5. The authors are encouraged to do some ablation studies to discuss the contribution of each different clustering algorithm.
6. The authors are proposed to compare the attention scores learned by the proposed method with vanilla graph attention and graph Transformer, to illustrate the effectiveness of the proposed method.


---
Reference:

- [1] Differentiable Clustering for Graph Attention.
- [2] Attention-based Graph Clustering Network with Dual Information Interaction.
- [3]. Transforming Graphs for Enhanced Attribute Clustering: An Innovative Graph Transformer-Based Method.

**Questions:**

Refer to "Weaknesses"

---

> ### Author Response · Authors · 2025-11-18
> **Author Rebuttal**
>
> Thank you for your valuable feedback! We address your concerns below.
>
> 1. You are right about the computational complexity. However, in most real-world graphs, the clusters found by classic community detection algorithms are significantly smaller than the size of the whole graph (i.e., even if a graph has a giant connected component, it will typically be separated into multiple clusters). Thus, while in the worst case our method’s complexity is similar to Global Graph Transformers (which are still efficient enough to be very popular), for most practical graphs it is much more efficient. As our experiments show, we are able to scale CLATT even to the huge pokec-regions dataset with more than 1.5 million nodes and more than 22 million edges, to which Global Graph Transformers do not scale. Please note that this datasets is orders of magnitude larger than graphs most popular in current graph ML research such as cora and citeseer.
>
> 2. Indeed, our method is designed for graphs that possess meaningful cluster structure. Graphs can be used to represent data from very different domains, and thus can exhibit a wide range of structural characteristics. Therefore, we do not expect any single method to work well across all possible graphs, and constructing counterexamples for any method including CLATT is possible. However, in our experiments, we demonstrate that CLATT improves performance on a wide range of real-world graphs. This is in line with classical observations from network science that many real-world social, information, and biological networks exhibit well-defined community structure. We discuss this in our Limitations section, where we specifically mention highly-regular grid-like graphs as an example of graphs without cluster structure. However, classical convolutional architectures are probably better suited for grid graphs anyway.
>
> 3. Inductive settings are challenging for all neural architectures for graphs. While clusters might be noticeably affected by adding new nodes and edges, so will be positional encodings for Global Graph Transformers. And, as recent work [4] shows, even classic MPNNs typically suffer in the inductive setting. Thus, most current graph machine learning works focus on the transductive setting, and adapting models to inductive settings is an open problem for all architectures.
>
> 4. Indeed, there are prior works bridging graph clustering and GNNs, however, they typically focus on using GNNs to cluster a graph (also referred to as graph pooling) often optimizing some version of soft modularity, while our work is to the best of our knowledge the first one to propose using graph clusterings computed with classic algorithms to create a new type of structure-aware graph attention and use it for standard node property prediction tasks.
>
>     Regarding more specifically the works mentioned by you:
>
>     [1] adds an auxiliary graph clustering objective (based on soft modularity) into the standard training pipeline of GAT, while the local attention mechanism of GAT stays unmodified.
>
>     [2] and [3] use attention-based neural models to cluster a graph, while we use a precomputed graph clustering to create a new attention type for node property prediction tasks.
>
> 5. We find that different clustering algorithms work better for different graphs, which is expected as the considered graphs strongly differ in their structure, and different clustering algorithms rely on different assumptions about the graph structure. However, for some graphs with particularly well-defined cluster structure such as city-roads-M/L or amazon-ratings all clustering methods work well. We have launched a set of experiments to demonstrate the effect from different clustering algorithms and will add it to the final version of the paper.
>
> 6. Thank you for this suggestion. We have added Appendix D with this analysis to the updated version of our paper. As expected, cluster attention typically works over longer distances than local (neighborhood) attention but shorter distances than global (full-graph) attention. Combined with its empirical benefits demonstrated in our experiments, this confirms that longer-range interactions than local message passing allows are beneficial, but Global Graph Transformers struggle to capture structurally meaningful dependencies due to a дфсл of structure-based inductive biases.
>
>
>
> [4] GraphLand: Evaluating Graph Machine Learning Models on Diverse Industrial Data (NeurIPS 2025)

---

### Author Response · Authors · 2025-12-02
**Rebuttal summary**

Dear Reviewers and Area Chair,

We thank the reviewers for their feedback on our work. We are glad that the reviewers appreciated the strong empirical performance of our method on diverse realistic graph datasets. Unfortunately, the new rules do not allow the reviewers to continue the discussion, however, we believe we were able to thoroughly address the reviewers’ concerns. Here we briefly summarise a couple points that appeared several times in the discussion.

First, some reviewers asked for more analysis of the proposed method’s inner workings. To address this, we have added Appendix D to the updated version of our paper in which we analyse the attention patterns of cluster attention and compare them to local (1-hop neighborhood) attention and global (all-to-all) attention. On the one hand, we show that cluster attention typically works at a significantly longer range than local 1-hop interactions of classic MPNNs, and often even at a longer range than the entire receptive field of multilayer MPNNs, showing that longer-range interactions than those that can be captured by MPNNs are useful. On the other hand, we show that cluster attention typically works at a smaller range than Global Graph Transformer attention, which, combined with CLATT’s empirical performance, suggests that there are useful mid-range interactions on which Global Graph Transformers fail to focus enough due to their lack of structure-based inductive biases. This confirms that cluster attention achieves its strong empirical performance due to reasons outlined in our motivation: by allowing capturing long-range interactions but still being strongly rooted in the graph structure and providing the valuable structure-based inductive biases.

Second, several reviewers had concerns about the novelty of our work compared to prior works bridging GNNs and graph clustering. We emphasize that, to the best of our knowledge, our work is the first to propose using graph clustering to define a new type of graph attention (which combines long-range interactions with strong structure-based inductive biases). In contrast, prior works combining graph clustering and GNNs typically focus on using GNNs to cluster a graph (also referred to as graph pooling), which is different from our work in both the considered task and the methods used. Some works also add virtual nodes or some variant of hierarchical interactions to GNNs based on graph clusters, but all these methods introduce a bottleneck that necessitates compressing all cluster information into a fixed-size vector. This problem is similar to the representation bottleneck in classic attentionless seq2seq models. In contrast, cluster attention allows unrestricted pairwise communications between all nodes in a cluster, which, as we show in our reply to Reviewer cWK1, results in achieving significantly better results in practice.

Sincerely,

The Authors

---

### Meta-Review · Area_Chair_Yfri · 2025-12-23

**Summary:**

This paper introduces Cluster Attention (CLATT), a method for aligning attention sparsity patterns in attention-based GNNs with clusters in a graph. Clusters are obtained from off-the-shelf graph clustering algorithms. CLATT shows improvements in predictive performance compared to baseline GNNs without this sparsity pattern.

Overall, the AC believes that this is a promising method that has the potential to become a go-to approach for graph machine learning. A related approach has been accepted concurrently at AAAI 2026 [1], further highlighting the timeliness of the proposed method. (Note that the authors are not expected to compare to this concurrent work, even though an earlier version of this work has been available publicly on OpenReview since September 2024, but in an unpublished state.)

To meet the bar for acceptance at ICLR, however, the paper needs some significant additional work in order to address reviewer feedback, but also to generally improve the quality of the work in the following areas:

1) Contextualization against the existing rich literature on hierarchical message passing, graph clustering in the context of GNNs, and GNNs with virtual nodes. For reference, the concurrent work [1] provides a reasonable literature overview, but further depth would be appreciated. 3 out of 4 reviewers highlight this issue.

2) Experimental analysis: Two aspects are lacking, a) detailed experimental comparison of clustering algorithms as basis for the approach, and b) an analysis of computational cost / overhead, FLOPs etc. — In their rebuttal, the authors mention a 20% wall-clock time overhead compared to one baseline in the response to reviewer Cdbo. So it appears the method is trading off additional computational overhead for a mild gain in predictive quality. It would be good to analyze how this compares to other approaches that make this trade-off.

3) Clarity of writing. The paper would overall benefit from higher quality of writing and exposition, as highlighted by the reviewers.

In summary, the paper does not yet meet the bar for ICLR. The approach is promising and the paper will likely face positive reception at a future venue after the authors address the reviewer feedback in detail.

[1] Choi et al., Are Graph Transformers Necessary? Efficient Long-Range Message Passing with Fractal Nodes in MPNNs, AAAI 2026

**Reviewer Concerns:**

Multiple reviewers mention a lack of theoretical analysis as a weakness of this paper. The authors have responded to this in the rebuttal and the AC sides with the authors here: since the algorithm makes no assumptions about the clustering algorithm used as a basis for clustering (it is compatible with a range of algorithms), a general theoretical analysis is out of scope.

The other points (1-3 above) have not been sufficiently addressed in the rebuttal.

**Reviewer Scores:**

The two weak reject voting reviewers could have likely changed their vote to weak accept. The AC believes the other two reviewers would have likely not changed their score: the weak accepting reviewer already assigned a relatively positive score despite listing a range of weaknesses, and the reject voting reviewers’ arguments have not been fully addressed (incl. no response to the requested experimental comparison of different clustering algorithms).

---

### Decision · Program_Chairs · 2026-01-26

Reject